# On Interaction Between Augmentations and Corruptions in Natural Corruption Robustness

**Eric Mintun**[*]
Facebook AI Research
mintun@fb.com

**Alexander Kirillov**
Facebook AI Research
akirillov@fb.com

**Saining Xie**
Facebook AI Research
s9xie@fb.com

## Abstract

Invariance to a broad array of image corruptions, such as warping, noise, or color shifts, is an important aspect of building robust models in computer vision. Recently, several new data augmentations have been proposed that significantly improve performance on ImageNet-C, a benchmark of such corruptions. However, there is still a lack of basic understanding on the relationship between data augmentations and test-time corruptions. To this end, we develop a feature space for image transforms, and then use a new measure in this space between augmentations and corruptions called the Minimal Sample Distance to demonstrate a strong correlation between similarity and performance. We then investigate recent data augmentations and observe a significant degradation in corruption robustness when the test-time corruptions are sampled to be perceptually dissimilar from ImageNet-C in this feature space. Our results suggest that test error can be improved by training on perceptually similar augmentations, and data augmentations may not generalize well beyond the existing benchmark. We hope our results and tools will allow for more robust progress towards improving robustness to image corruptions. We provide code at `https://github.com/facebookresearch/augmentation-corruption`.

## 1   Introduction

Robustness to distribution shift, *i.e.* when the train and test distributions differ, is an important feature of practical machine learning models. Among many forms of distribution shift, one particularly relevant category for computer vision are image corruptions. For example, test data may come from sources that differ from the training set in terms of lighting, camera quality, or other features. Post-processing transforms, such as photo touch-up, image filters, or compression effects are commonplace in real-world data. Models developed using clean, undistorted inputs typically perform dramatically worse when confronted with these sorts of image corruptions [8, 13]. The subject of corruption robustness has a long history in computer vision [1, 6, 28] and recently has been studied actively with the release of benchmark datasets such as ImageNet-C [13].

One particular property of image corruptions is that they are low-level distortions in nature. Corruptions are transformations of an image that affect structural information such as colors, textures, or geometry [5] and are typically free of high-level semantics. Therefore, it is natural to expect that *data augmentation* techniques, which expand the training set with random low-level transformations, can help learn robust models. Indeed, data augmentation has become a central technique in several recent methods [14, 20, 25] that achieve large improvements on ImageNet-C and related benchmarks.

One caveat for data augmentation based approaches is the test corruptions are expected to be *unknown* at training time. If the corruptions are known, they may simply be applied to the training set as data augmentations to trivially adapt to the test distribution. Instead, an ideal robust model needs to be

---

[*]This work completed as part of the Facebook AI residency program.

35th Conference on Neural Information Processing Systems (NeurIPS 2021).

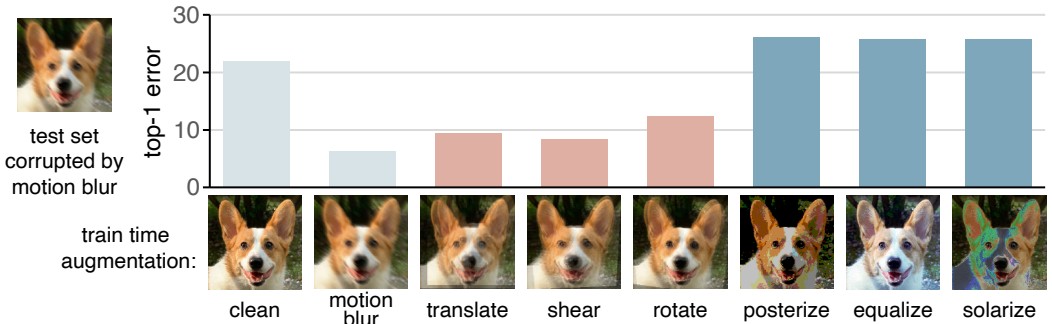

Figure 1: **A toy experiment.** We train multiple models on CIFAR-10 [17] using different augmentation schemes. Each scheme is based on a single basic image transformation type and enhanced by overlaying random instantiations of the transformation for each input image following Hendrycks et al. [14]. We compare these models on the CIFAR-10 test set corrupted by the motion blur, a corruption used in the ImageNet-C corruption benchmark [13]. None of the augmentation schemes contains motion blur; however, the models trained with geometric-based augmentations significantly outperform the baseline model trained on the clean images while color-based augmentations show no gains. We note the geometric augmentations can produce a result visually similar to a blur by overlaying copies of shifted images[3].

robust to *any* valid corruption, including ones unseen in any previous benchmark. Of course, in practice the robustness of a model can only be evaluated approximately by measuring its corruption error on a representative corruption benchmark. To avoid trivial adaptation to the benchmark, recent works manually exclude test corruptions from the training augmentations. However, with a toy experiment presented in Figure 1, we argue that this strategy alone might not be enough and that visually similar augmentation outputs and test corruptions can lead to significant benchmark improvements even if the exact corruption transformations are excluded.

This observation raises two important questions. One, *how exactly does the similarity between train time augmentations and corruptions of the test set affect the error?* And two, if the gains are due to the similarity, they may not translate into better robustness to other possible corruptions, so *how well will data augmentations generalize beyond a given benchmark?* In this work, we take a step towards answering these questions, with the goal of better understanding the relationship between data augmentation and test-time corruptions. Using a feature space on image transforms and a new measure called Minimal Sample Distance (MSD) on this space, we are able to quantify the distance between augmentation schemes and classes of corruption transformation. With our approach, we empirically show an intuitive yet surprisingly overlooked finding:

*Augmentation-corruption perceptual similarity is a strong predictor of corruption error.*

Based on this finding, we perform additional experiments to show that data augmentation aids corruption robustness by increasing perceptual similarity between a (possibly small) fraction of the training data and the test set. To further support our claims, we introduce a set of new corruptions, called CIFAR/ImageNet-$\overline{\text{C}}$, to test the degree to which common data augmentation methods generalize from the original CIFAR/ImageNet-C. To choose these corruptions, we expand the set of natural corruptions and sample new corruptions that are far away from CIFAR/ImageNet-C in our feature space for measuring perceptual similarity. We then demonstrate that augmentation schemes designed specifically to improve robustness show significantly degraded performance on CIFAR/ImageNet-$\overline{\text{C}}$. Some augmentation schemes still show some improvement over baseline, which suggests meaningful progress towards general corruption robustness is being made, but different augmentation schemes exhibit different degrees of generalization capability. As an implication, caution is needed for fair robustness evaluations when additional data augmentation is introduced.

These results suggest a major challenge that is often overlooked in the study of corruption robustness: *generalization is often poor*. Since perceptual similarity can predict performance, for any fixed finite set of test corruptions, improvements on that set may generalize poorly to dissimilar corruptions. We

---

[3]Example transforms are for illustrative purpose only and are exaggerated. Base image © Sehee Park.

hope that these results, tools, and benchmarks will help researchers better understand *why* a given augmentation scheme has good corruption error and whether it should be expected to generalize to dissimilar corruptions. On the positive side, our experiments show that *generalization does emerge* among perceptually similar transforms, and that only a *small fraction* of sampled augmentations need to be similar to a given corruption. Section 6 discusses these points in more depth.

## 2 Related Work

**Corruption robustness benchmarks and analysis.** ImageNet-C [13] is a corruption dataset often used as a benchmark in robustness studies. Other corruption datasets [15, 27] collect corrupted images from real world sources and thus have a mixture of semantic distribution shifts and perceptual transforms. Corruption robustness differs from adversarial robustness [31], which seeks invariance to small, worst case distortions. One notable difference is that improving corruption robustness often slightly improves regular test error, instead of harming it. Yin et al. [38] analyzes corruption robustness in the context of transforms' frequency spectra; this can also influence corruption error independently from perceptual similarity. Here we study the relationship between augmentations and corruptions more generically, and explore the relationship between perceptual similarity and generalization to new corruptions. Dao et al. [3] and Wu et al. [36] study the theory of data augmentation for regular test error. Hendrycks et al. [15] and Taori et al. [33] study how the performance on synthetic corruption transforms generalizes to performance on corruption datasets collected from the real world. Here we do not address this issue directly but touch upon it in the discussion.

**Improving corruption robustness.** Data augmentations designed to improve robustness include AugMix [14], which composites common image transforms, Patch Gaussian [20], which applies Gaussian noise in square patches, and ANT [25], which augments with an adversarially learned noise distribution. AutoAugment [2] learns augmentation policies that optimize clean error but has since been shown to improve corruption error [38]. Mixup [40] can improve robustness [18], but its label augmentation complicates the dependence on image augmentation. Stylized-ImageNet [9], which applies style transfer to input images, can also improve robustness. DeepAugment [15], which applies augmentations to a deep representation of an image, can also give large improvements in robustness. Noisy Student [37] and Assemble-ResNet [18] combine data augmentation with new models and training procedures and greatly enhance corruption robustness. In addition to training-time methods, there are approaches that adapt to unseen corruptions at test time, e.g. using self-supervised tasks [30], entropy minimization [35], or with a focus on privacy and data transmission efficiency [19]. While we do not directly address these approaches here, our methods potentially provide tools that could be used to measure shifting distributions in an online regime.

## 3 Perceptual similarity for augmentations and corruptions

First, we study the importance of similarity between augmentations and corruptions for improving performance on those corruptions. To do so, we need a means to compare augmentations and corruptions. Both types of transforms are perceptual in nature, meaning they affect low-level image structure while leaving high-level semantic information intact, so we expect a good distance to be a measure of *perceptual similarity*. Then, we need to find the appropriate measure of distance between the augmentation and corruption *distributions*. We will argue below that distributional equivalence is not appropriate in the context of corruption robustness, and instead introduce the *minimal sample distance*, a simple measure that does capture a relevant sense of distribution distance.

**Measuring similarity between perceptual transforms.** We define a perceptual transform as a transform that acts on low-level image structure but not high-level semantic information. As such, we expect two transforms should be similar if their actions on this low-level structure are similar, independent of algorithmic or per-pixel differences between them. A closely related, well-studied problem is the perceptual similarity between *images*. A common approach is to train a neural network on a classification task and use intermediate layers as a feature space for measuring distances [42]. We adapt this idea to obtain a feature space for measuring distances between perceptual transforms.

We start with a feature extractor for images, which we call $\hat{f}(x)$. To train the model from which we will extract features, we assume access to a dataset $\mathbb{D}$ of image label pairs $(x, y)$ associated with a

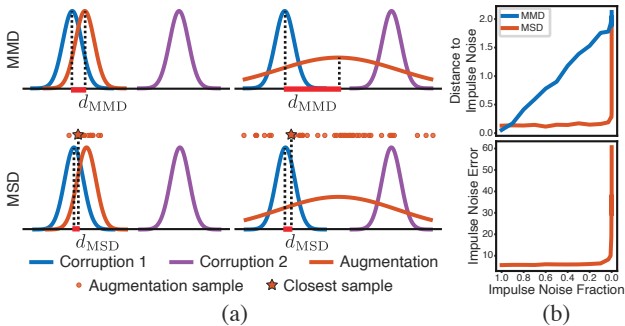

Figure 2: (a) Schematic comparison of MMD to MSD. MMD measures the distance between distribution centers and is only small if the augmentation overlaps with a corruption. MSD measures to the nearest sampled point in the set of samples (marked by a star) and is small even for broad distributions that overlap with multiple corruptions. (b) We test on images corrupted with *impulse noise*, and train on images augmented with a mixture of *impulse noise* and *motion blur*. As the mixing fraction of *impulse noise* decreases, MMD between the augmentation and corruption grows linearly while MSD and error stay low until nearly 0% mixing fraction.

classification task. The model should be trained using only default data augmentation for the task in question so that the feature extractor is independent of the transforms we will use it to study. In order to obtain a very simple measure, we use just the last hidden layer of the network as a feature space.

A perceptual transform $t(x)$ may be encoded by applying it to all images in $\mathbb{D}$, encoding the transformed images, and averaging the features over these images. For efficiency, we find it sufficient to average over only a randomly sampled subset of images $\mathbb{D}_S$ in $\mathbb{D}$. In Section 4.1 we discuss the size of $\mathbb{D}_S$. The random choice of images is a property of the feature extractor, and so remains fixed when encoding multiple transforms. This reduces variance when computing distances between two transforms. The transform feature extractor is given by $f(t) = \mathbb{E}_{x \in \mathbb{D}_S}[\hat{f}(t(x)) - \hat{f}(x)]$. The *perceptual similarity* between an augmentation and a corruption can be taken as the $L_2$ distance on this feature space $f$.

**Minimal sample distance.**    We now seek to compare the distribution of an augmentation scheme $p_a$ to a distribution of a corruption benchmark $p_c$. If the goal was to optimize error on a *known* corruption distribution, exact equivalence of distributions is the correct measure to minimize. But since the goal is robustness to general, *unknown* corruption distributions, a good augmentation scheme should be equivalent to no single corruption distribution.

To illustrate this behavior, consider a toy problem where we have access to the corruption transforms at training time. A very rough, necessary-but-insufficient measure of distributional similarity is $d_{\mathrm{MMD}}(p_a, p_c) = ||\mathbb{E}_{a \sim p_a}[f(a)] - \mathbb{E}_{c \sim p_c}[f(c)]||$. This is the maximal mean discrepancy on a fixed, finite feature space, so for brevity we will refer to it as MMD. We still employ the featurization $f(t)$, since we are comparing transforms and not images, unlike in typical domain adaptation. Consider two corruption distributions, here *impulse noise* and *motion blur*, and an augmentation scheme that is a mixture of the two corruption distributions. Figure 2b shows MMD between the augmentation and *impulse noise* corruption scales linearly with mixing fraction, but error on *impulse noise* remains low until the mixing fraction is almost 0% impulse noise. This implies distributional similarity is a poor predictor of corruption error. Indeed, low $d_{\mathrm{MMD}}$ with any one corruption distribution suggests the augmentation overlaps it significantly, so the augmentation is unlikely to aid dissimilar corruptions.

Our expectation for the behavior of the error in Figure 2b is that networks can often successfully memorize rare examples seen during training, so that only a very small fraction of sampled images need *impulse noise* augmentations to perform well on *impulse noise* corruptions. An appropriate distance should then measure how close augmentation samples can come to the corruption distribution, even if the density of those samples is low. We thus propose a very simple measure called *minimal sample distance (MSD)*, which is just the perceptual similarity between an average corruption and the closest augmentation from a finite set of samples $\mathbb{A} \sim p_a$:

$$d_{\mathrm{MSD}}(p_a, p_c) = \min_{a \in \mathbb{A} \sim p_a} ||f(a) - \mathbb{E}_{c \sim p_c}[f(c)]|| \,. \tag{1}$$

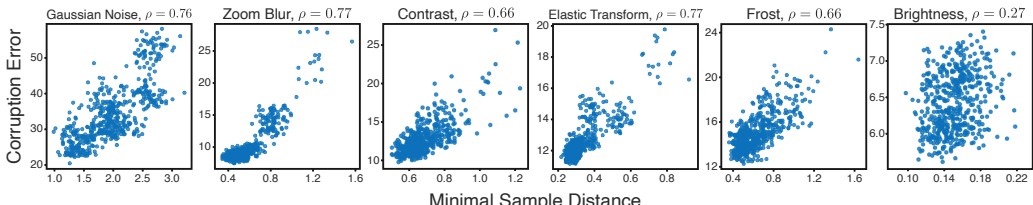

Figure 3: Example relationships between MSD and corruption error. $\rho$ is the Spearman rank correlation. MSD correlates well with error across all four categories of corruption in CIFAR-10-C. For completeness, we also show *brightness*, a negative example where correlation is poor.

A schematic comparison of MMD and MSD is shown in Figure 2a. While both MMD and MSD are small for an augmentation scheme that is distributionally similar to a corruption distribution, only MSD remains small for a broad distribution that occasionally produces samples near multiple corruption distributions. Figure 2b shows MSD, like test error, is small for most mixing fractions in the toy problem described above. Note the measure's need to accommodate robustness to general, unknown corruption distributions has led it to be asymmetric, so it differs from more formal distance metrics that may be used to predict generalization error, such as the Wasserstein distance [43].

## 4 Perceptual similarity is predictive of corruption error

We are now equipped to measure how important this augmentation-corruption similarity is for corruption error. For a large number of augmentation schemes, we will measure both the MSD to a corruption distribution and the corruption error of a model trained with that scheme. We will find a correlation between MSD and corruption error, which provides evidence that networks generalize across perceptually similar transforms. Then, we will calculate MSD for augmentation schemes in the literature that have been shown to improve error on corruption benchmarks. We will find a correlation between MSD and error here as well, suggesting their success is in part explained by their perceptual similarity to the benchmark. This implies there may be a risk of poor generalization to different benchmarks, since we would not expect this improvement to transfer to a dissimilar corruption.

### 4.1 Experimental setup

**Corruptions.** We use CIFAR-10-C [13], which is a common benchmark used for studying corruption robustness. It consists of 15 corruptions, each further split into five different severities of transformation, applied to the CIFAR-10 test set. The 15 corruptions fall into four categories: per-pixel noise, blurring, synthetic weather effects, and digital transforms. We treat each corruption at each severity as a separate distribution for the sake of calculating MSD and error; however, for simplicity we average errors and distances over severity to present a single result per corruption.

**Space of augmentation schemes.** To build each sampled augmentation transform, we will composite a set of base augmentations. For base augmentations, we consider the nine common image transforms used in Hendrycks et al. [14]. There are five geometric transforms and four color transforms. By taking all subsets of these base augmentations, we obtain $2^9 = 512$ unique augmentation schemes, collectively called the *augmentation powerset*. Also following Hendrycks et al. [14], we composite transforms in two ways: by applying one after another, or by applying them to copies of the image and then linearly superimposing the results. Examples of both augmentations and corruptions are provided in Appendix F.

**Computing similarity and corruption error.** A WideResNet-40-2 [39] model is pre-trained on CIFAR-10 using default augmentation and training parameters from Hendrycks et al. [14]. WideResNet is a common baseline model used when studying data augmentation on CIFAR-10 [2, 14, 40]. Its last hidden layer is used as the feature space. For MSD, we average over 100 images, 100 corruptions, and minimize over 100k augmentations. With this number of corruptions and images, we find that the average standard deviation in distance between an augmentation and the averaged corruptions is roughly five percent of the mean, which is smaller than the typical feature in our results found below, given in Figure 3. We also find that using VGG [29] instead of WideResNet for the feature extractor

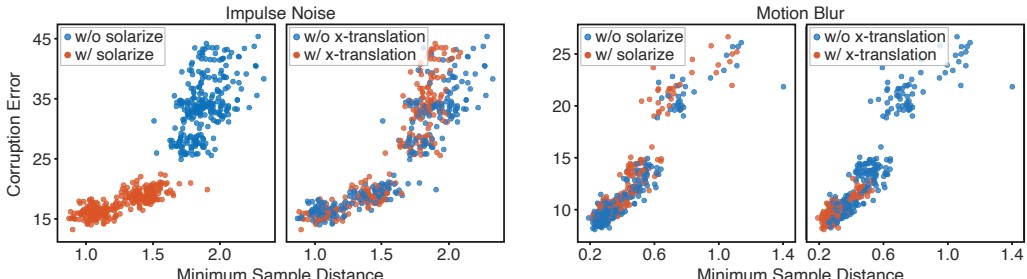

Figure 4: Example relationships between base augmentations and corruptions. Including *solarize* reduces MSD on the perceptually similar *impulse noise* corruption. Including *x translation* reduces MSD on the perceptually similar *motion blur* corruption. MSD is not decreased for dissimilar augmentation-corruption pairs.

gives similar results. Details for these calculations are in Appendix C. Images for calculating MSD are from the training set and do not have default training augmentation. A WideResNet-40-2 with the same training parameters is used for corruption error evaluation.

### 4.2 Analysis

**MSD correlates with corruption error.** First, we establish the correlation between MSD and corruption error on the augmentation powerset. MSD shows strong correlation with corruption error across corruptions types in all four categories of CIFAR-10-C, and for a large majority of CIFAR-10-C corruptions in general: 12 of 15 have Spearman rank correlation greater than 0.6. Figure 3 shows the relationship between distance and corruption error on six example corruptions, including one negative example for which correlation is low. A complete set of plots is below in Figure 5. This corruption, *brightness*, may give poor results because it is a single low-level image statistic that can vary significantly from image to image, and thus may not be well represented by our feature extractor. Appendix B has a few supplemental experiments. First, we we confirm MMD correlates poorly with corruption error, as expected. In particular, we expect broad augmentation schemes produce samples similar to a larger set of corruptions, leading to both lower MSD and lower corruption error but higher MMD. Second, we repeat our experiment but do not train on the augmentations, instead only adapting the batch norm statistics of a pre-trained model to them. We still find a strong correlation, suggesting our methods are compatible with the results of Schneider et al. [26], which shows such an adaptation of the batch norm statistics to a corruption can improve corruption error.

**An example of perceptual similarity.** Here we illustrate the perceptual nature of the similarity measure, using an example with two base augmentations and two corruptions. The augmentation *solarize* and the corruption *impulse noise* both insert bright pixels into the image, though in different ways. Linear superpositions of the augmentation *x translation* are visually similar to a blur, such as the corruption *motion blur*. Figure 4 shows MSD vs error where augmentation schemes that include *solarize* and *x translation* are colored. It is clear that including an augmentation greatly decreases MSD to its perceptually similar corruption, while having little effect on MSD to its perceptually dissimilar corruption.

**MSD and corruption error in real augmentation methods.** The augmentation powerset may be used as a baseline for comparing real data augmentation schemes. Figure 5 shows MSD-error correlations for Patch Gaussian [20], AutoAugment [2], and Augmix [14], along with the cloud of augmentation powerset points for all 15 CIFAR-10-C corruptions. The real augmentation schemes follow the same general trend that lower error predicts lower MSD. A few intuitive correlations are also captured in Figure 5. Patch Gaussian has low MSD to noise corruptions. AutoAugment, which contains contrast and Gaussian blurring augmentations in its sub-policies, has low MSD with *contrast* and *defocus blur*. A negative example is *fog*, on which MSD to AutoAugment is not predictive of corruption error.

This correlation suggests generalization may be poor beyond an existing benchmark, since an augmentation scheme may be perceptually similar to one benchmark but not another. For augmentations and corruptions that are explicitly the same, such as *contrast* in AutoAugment and ImageNet-C, this

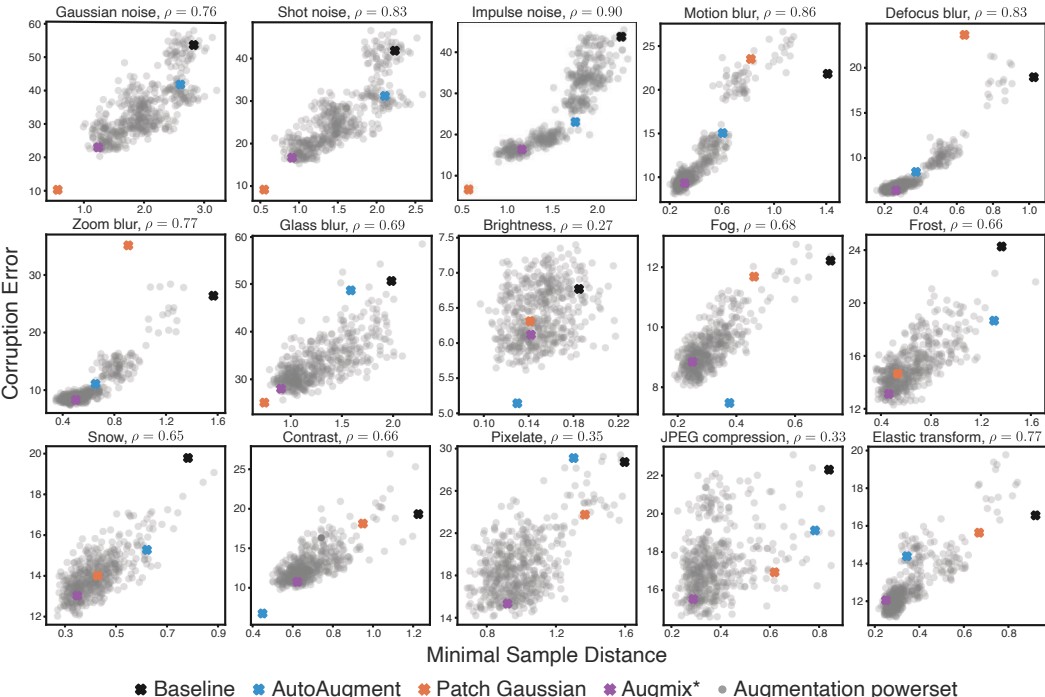

Figure 5: Correlations for augmentation schemes from the literature. Patch Gaussian is similar to noise, while AutoAugment is similar to contrast and blur, as expected from their formulation. Glass blur acts more like a noise corruption than a blur for these augmentation schemes, likely because it randomly permutes pixels. As a negative example, MSD does not correlate well with error for AutoAugment on *fog*. [*]AugMix here refers to just the augmentation distribution in Hendrycks et al. [14], not the proposed Jensen-Shannon divergence loss.

is typically accounted for by removing such transforms from the augmentation scheme when testing corruption robustness[4]. But in addition to these explicit similarities, Figure 5 shows quantitatively that perceptual similarity between non-identical augmentations and corruptions is also strongly predictive of corruption error. This includes possibly unexpected similarities, such as between Patch Gaussian and *glass blur*, which introduces random pixel-level permutations as noise. This suggests that perceptually similar augmentations and corruptions should be treated with the same care as identical transforms. In particular, tools such as MSD help us determine *why* an augmentation scheme improves corruption error, so we can better understand if new methods will generalize beyond their tested benchmarks. Next we test this generalization by finding corruptions dissimilar to ImageNet-C.

## 5   ImageNet-$\overline{\text{C}}$: benchmarking with dissimilar corruptions

We now introduce a set of corruptions, called ImageNet-$\overline{\text{C}}$, that are perceptually dissimilar to ImageNet-C in our transform feature space, and we will show that several augmentation schemes have degraded performance on the new dataset. We emphasize that the dataset selection method uses only default data augmentation and was fixed before we looked at the results for different augmentations, so we are not adversarially selecting against the tested augmentation schemes.

**Dataset construction.**   Here we present an overview of the dataset construction method. We build 30 new corruptions in 10 severities, from which the 10 most dissimilar corruptions will be chosen. We adapt common filters and noise distributions available online [10, 16] to produce human interpretable images. The transforms include warps, blurs, color distortions, noise additions, and obscuring effects. Examples of the new corruptions and exact details of the construction method are provided in Appendices D and F.

---

[4]For this analysis, we wish to treat explicit transform similarity and perceptual transform similarity on the same footing, so we choose not to remove these overlapping augmentations.

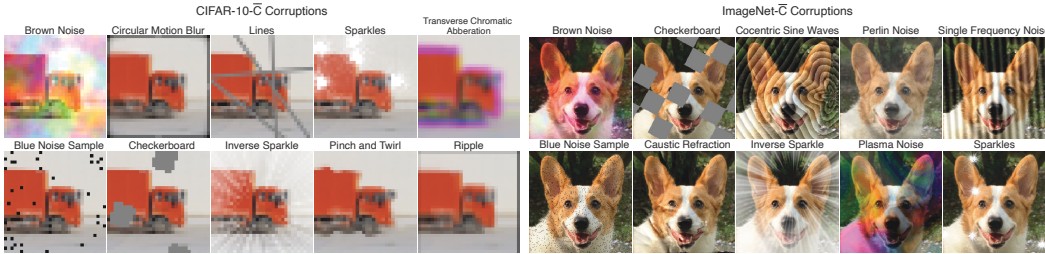

Figure 6: Example CIFAR-10-C̄ and ImageNet-C̄ corruptions. While still human interpretable, new corruptions are sampled to be dissimilar from CIFAR-10/ImageNet-C. Base images © Sehee Park and Chenxu Han.

To assure that the new dataset is no harder than ImageNet-C, we restrict the average corruption error of the new dataset to be similar to that of ImageNet-C for default augmentation. We then generate many potential datasets and measure the average shift in distance to ImageNet-C that each corruption contributes. Note that while MSD is a measure between augmentations and corruptions, here we are comparing corruptions to other corruptions and thus use MMD in our transform feature space. ImageNet-C̄ then consists of the 10 corruptions types with the largest average shift in distance. Like ImageNet-C, each has five different severities, with severities chosen so that the average error matches ImageNet-C for default augmentation. Example transforms from ImageNet-C̄ and CIFAR-10-C̄ are shown in Figure 6. This procedure in our feature space produces corruptions intuitively dissimilar from ImageNet-C and CIFAR-10-C.

**Results.** We test AutoAugment [2], Patch Gaussian [20], AugMix [14], ANT$^{3x3}$ [25], Stylized-ImageNet [9], and DeepAugment [15] on our new datasets and show results in Table 1. CIFAR-10 models are WideResNet-40-2 with training parameters from Hendrycks et al. [14]. ImageNet [4] models are ResNet-50 [12] with training parameters from Goyal et al. [11]. Stylized-ImageNet is trained jointly with ImageNet for half the epochs and starts from a model pre-trained on ImageNet, following Geirhos et al. [9]. Models use default data augmentation as well as the augmentation being tested, except ImageNet color jittering is not used. All corruptions are applied in-memory instead of loaded from a compressed file; this can affect results especially on high frequency corruptions.

Since Section 4 suggests several augmentation schemes are perceptually similar to ImageNet-C corruptions, we might expect these methods to have worse error on the new corruptions. Indeed, every augmentation scheme performs worse. Different augmentation schemes also degrade by significantly different amounts, from +0.7% for AutoAugment to +7.3% for PatchGaussian, which changes their ranking by corruption error and leads to inconsistency of generalization. In Table 2, we compare performance on several robust models[7, 21, 22, 32, 34, 37, 41] that are not primarily augmentation-based and see no similar pattern of degradation, further suggesting that augmentation-corruption dissimilarity is the cause of the higher error.

Errors of individual corruptions in ImageNet-C̄ are also revealing. For all augmentation schemes, there is significant improvement on *blue sample noise*[5] but little improvement on *sparkles* or *inverse sparkles*. Only AutoAugment does well on *checkerboard*, perhaps because only AutoAugment's geometric transforms produce empty space, similar to *checkerboard*'s occluded regions. These examples suggest a slightly different benchmark could yield significantly different results. Indeed, for a hypothetical benchmark that excluded *blue sample noise* and *checkerboard*, AutoAugment and Patch Gaussian have 57.3% and 57.2% error respectively, little better than baseline of 57.4%. AugMix fairs only a little better with 54.3% error. Even DeepAugment+AugMix, which is in general a strong augmentation scheme, shows a big discrepancy in performance across different corruptions, improving *single frequency noise* by 31%, but *inverse sparkles* by only 2.3%. Generalization to dissimilar corruptions is thus both inconsistent and typically quite poor. Single benchmarks and aggregate corruption scores are likely not enough for careful evaluation of robustness to unknown corruptions, and it is important to study why proposed augmentations succeed to better understand how well they might generalize.

---

[5]This corruption is conceptually similar with *impulse noise* but also gives a large distance; this may be a failure mode of our measure, maybe since *impulse noise* has bright pixels and *blue noise sample* has dark pixels.

Table 1: Test error for several data augmentation methods on CIFAR-10-C̄ and ImageNet-10-C̄, for which every method performs worse than on ImageNet-C or CIFAR-10-C. The increase in error differs significantly between different augmentation methods. Descriptions of the abbreviations and standard deviations for individual corruptions are in Appendix D. 'Baseline' refers to default augmentation only. Averages are over five runs for ImageNet and ten for CIFAR-10. *ANT, DeepAugment(DA) and DeepAugment+AugMix (DA+AM) use the pre-trained model provided with the associated papers and have different training parameters.

| | **IN-C** | **IN-C̄** | | **ImageNet-C̄ Corruptions** | | | | | | | | | |
| Aug | Err | Err | ΔIN-C | BSmpl | Plsm | Ckbd | CSin | SFrq | Brwn | Prln | Sprk | ISprk | Rfrac |
|---|---|---|---|---|---|---|---|---|---|---|---|---|---|
| Baseline | $58.1_{\pm0.4}$ | $57.7_{\pm0.2}$ | -0.4 | 68.6 | 71.7 | 49.4 | 84.7 | 79.0 | 37.5 | 34.3 | 32.4 | 76.7 | 42.8 |
| AA | $55.0_{\pm0.2}$ | $55.7_{\pm0.3}$ | +0.7 | 54.8 | 68.3 | 43.8 | 86.5 | 78.8 | 34.5 | 33.8 | 36.1 | 77.1 | 43.8 |
| SIN | $52.4_{\pm0.1}$ | $55.8_{\pm0.3}$ | +3.4 | 54.7 | 69.8 | 52.8 | 79.6 | 69.2 | 37.8 | 35.3 | 37.0 | 77.3 | 44.1 |
| AugMix | $49.2_{\pm0.7}$ | $52.4_{\pm0.2}$ | +3.2 | 43.2 | 72.2 | 46.1 | 76.3 | 67.4 | 38.8 | 32.4 | 32.3 | 76.4 | 39.2 |
| PG | $49.3_{\pm0.2}$ | $56.6_{\pm0.4}$ | +7.3 | 60.3 | 74.1 | 48.5 | 82.1 | 76.7 | 38.9 | 34.6 | 32.1 | 76.5 | 42.1 |
| ANT* | 48.8 | 53.9 | +5.1 | 35.8 | 75.5 | 56.9 | 76.4 | 63.7 | 41.0 | 35.2 | 35.0 | 76.1 | 43.3 |
| DA* | 46.6 | 51.0 | +4.4 | 41.7 | 73.3 | 53.9 | 74.6 | 50.9 | 37.2 | 30.3 | 32.9 | 74.7 | 40.9 |
| DA+AM* | 41.0 | 48.3 | +7.3 | 34.9 | 67.9 | 49.8 | 69.7 | 48.0 | 35.2 | 30.6 | 32.9 | 74.3 | 39.8 |
| | **C10-C** | **C10-C̄** | | **CIFAR-10-C̄ Corruptions** | | | | | | | | | |
| Aug | Err | Err | ΔC10-C | BSmpl | Brwn | Ckbd | CBlur | ISprk | Line | P&T | Rppl | Sprk | TCA |
| Baseline | $27.0_{\pm0.6}$ | $27.1_{\pm0.5}$ | +0.1 | 42.9 | 27.2 | 23.3 | 11.8 | 43.3 | 26.2 | 11.3 | 21.6 | 21.0 | 42.9 |
| AA | $19.4_{\pm0.2}$ | $21.0_{\pm0.4}$ | +1.6 | 17.7 | 17.5 | 17.6 | 9.5 | 40.4 | 23.6 | 10.7 | 23.5 | 17.5 | 31.8 |
| AugMix | $11.1_{\pm0.2}$ | $16.0_{\pm0.3}$ | +5.9 | 9.8 | 27.8 | 13.4 | 5.9 | 30.3 | 18.0 | 8.3 | 12.1 | 15.5 | 19.2 |
| PG | $17.0_{\pm0.3}$ | $23.8_{\pm0.5}$ | +6.8 | 9.0 | 30.1 | 21.6 | 12.8 | 35.4 | 20.6 | 8.8 | 21.5 | 19.3 | 59.5 |

Table 2: Comparison of errors on ImageNet-C and ImageNet-C̄ for several robust models: WSL (weakly supervised ResNeXt-101-32x8d [21, 22]), EN (EfficientNet-B0 [32]), NS (Noisy Student EN-B0 [37]), ViT-S (Transformer [7, 34]), ResNeSt (ResNeSt-50d, [41]), using pre-trained models provided with the respective papers. These models do not rely primarily on data augmentation to be robust, and there is no consistent degradation on ImageNet-C̄. This is additional evidence that the worse performance in Table 1 does not occur because ImageNet-C̄ is harder generally.

| | WSL | EN | NS | ViT-S | ResNeSt |
|---|---|---|---|---|---|
| IN-C Err | 38.1 | 55.7 | 52.1 | 44.5 | 44.4 |
| IN-C̄ Err | 39.2 | 53.4 | 52.2 | 41.1 | 41.6 |

It may be surprising that Stylized-ImageNet also degrades, given that it is intuitively very different from every corruption. While our measure works for augmentations, it does not cover all possible methods that improve robustness, such as more complicated algorithms like Stylized-ImageNet. Stylized-ImageNet degradation may be due to other reasons. For instance, it primarily augments texture information and may help mostly with higher frequency corruptions, as can be seen by its improvement on *single frequency noise* and *cocentric sine waves*; ImageNet-C̄ has fewer such corruptions than ImageNet-C. ImageNet-C̄ is thus a useful tool for understanding the interaction between training procedure and corruption distribution, even beyond perceptual similarity.

Nevertheless, note that it is the intuitively broader augmentation schemes, such as AutoAugment, AugMix, Stylized-ImageNet, and DeepAugment that generalize better to ImageNet-C̄. The importance of breadth has also been explored elsewhere[15, 38], but in the previous sections we have provided new quantitative evidence for *why* this may be true: broad augmentation schemes may be perceptually similar to more types of corruptions, and thus more likely to be perceptually similar to a new corruption. Moreover, AugMix and DeepAugment still improve over baseline on ImageNet-C̄, so there is reason to be optimistic that robustness to unknown corruptions is an achievable goal, as long as evaluation is treated carefully.

## 6  Discussion

***Societal Impact.*** Our method for finding dissimilar corruptions could in principle be used to adversarially attack computer vision systems, such as those in content moderation or self-driving cars. Moreover, our ultimate goal is to help improve robustness in computer vision, and such robust

systems may be used in detrimentals ways, for example in autonomous weapons or surveillance. However, we expect better evaluation of robust models to have definite benefits as well. In the long run, such an understanding should help defend against adversarial attacks. Our tools could also be used to challenge purportedly robust systems that are actually dangerously unreliable, such as an autonomous driving system that is robust to common corruption benchmarks yet fails to be robust to a dissimilar but important corruption, e.g., maybe glare. For instance, is the model employing data augmentation that is perceptually similar to the corruptions being used to report good robustness? Is the set of validation corruptions sufficiently broad that we would expect reasonable generalization to an unseen corruption? If we generate a dissimilar set of corruptions using the procedure we develop here, does the model still perform well on the new corruptions? Quantitative ways to answer these questions may provide a means to verify the robust performance of a model before it encounters and potentially fails on a critical, previously unseen corruption.

***Corruption robustness as a secondary learning task.*** We have provided evidence that data augmentation may not generalize well beyond a given corruption benchmark. To explore this further, consider an analogy to a regular learning problem. We may think of corruption robustness in the presence of data augmentation as a sort of secondary task layered on the primary classification task: the set of data augmentations is the training set, the set of corruptions is the test set, and the goal is to achieve invariance of the underlying primary task. In this language, the 'datasets' involved are quite small: ImageNet-C has only 15 corruption types, and several augmentation schemes composite only around 10 basic transforms. In this case, standard machine learning practice would dictate a training/validation/test set split; it is only the size and breadth of modern vision datasets that has allowed this to be neglected in certain cases recently. But the effective dataset size of a corruption robustness problem is tiny, so having a held-out test set seems necessary. To emphasize, this is not a test set of the underlying classification task, for which generalization has been studied by Recht et al. [23, 24]. Instead, it is a test set of corruption transforms themselves. This means there would be validation/test split of dissimilar transformations, both applied to the ImageNet validation set[6].

***Real-world corruption robustness.*** Recently, Hendrycks et al. [15] and Taori et al. [33] study how performance on corruption transforms generalizes to real-world corruptions and come to conflicting conclusions. Though we do not study real-world corruptions, we have proposed a mechanism that may explain the conflict: performance will generalize between transforms and real-world corruptions if they are perceptually similar, but will likely not if they are dissimilar. Since Hendrycks et al. [15] and Taori et al. [33] draw on different real-world and synthetic corruptions, it may be that the perceptual similarity between datasets differs in the two analyses. This also suggests a way to find additional corruption transforms that correlate with real-world corruptions: transforms should be sought that have maximal perceptual similarity with real-world corruptions.

***Generalization does occur.*** We have encountered two features of data augmentation that may explain why it can be such a powerful tool for corruption robustness, despite the issues discussed above. First, within a class of perceptually similar transforms, generalization does occur. This means each simple data augmentation may confer robustness to many complicated corruptions, as long as they share perceptual similarity. Second, dissimilar augmentations in an augmentation scheme often causes little to no loss in performance, as long as a similar augmentation is also present. We briefly study this in Appendix A by demonstrating that adding many dissimilar augmentations increases error much less than adding a few similar augmentations decreases it. These two features suggest broad augmentation schemes with many dissimilar augmentations may confer robustness to a large class of unknown corruptions. More generally, we think data augmentation is a promising direction of study for corruption robustness, as long as significant care is taken in evaluation.

## Acknowledgements and Funding Disclosure

Eric Mintun would like to thank Matthew Leavitt, Sho Yaida, and Achal Dave for discussions during the development of this work. Additionally, he would like to acknowledge the Facebook AI residency program for providing excellent training and support in AI research. The authors received no external funding and have no competing interests.

---

[6]The validation set provided in Hendrycks & Dietterich [13] consists of perceptually similar transforms to ImageNet-C and would not be expected to work well for the validation discussed here.

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
