# Supplemental Material: On Interaction Between Augmentations and Corruptions in Natural Corruption Robustness

**Eric Mintun**[*]
Facebook AI Research
mintun@fb.com

**Alexander Kirillov**
Facebook AI Research
akirillov@fb.com

**Saining Xie**
Facebook AI Research
s9xie@fb.com

## A  Sampling similar augmentations more frequently gives minor performance improvements

Here we describe an alternative experiment that shows how the introduction of dissimilar augmentations affects corruption error. For a broad data augmentation scheme that provides robustness to many dissimilar corruptions, each corruption may only have a similar augmentation sampled some small fraction of the time. This small fraction of samples must be sufficient to yield good performance on each corruption to obtain robustness overall. We expect that this should be the case, since neural networks are often good at memorizing rare examples. Additionally, the toy problem in Figure 2 of Section 3 suggests that a large fraction of sampled augmentations may be dissimilar without significant loss in corruption error. Here we show the effect using a real augmentation scheme.

We consider performance on CIFAR-10-C when training with AugMix augmentations (we do not use their Jensen-Shannon divergence loss, which gives additional improvements). However, instead of sampling directly from the AugMix distribution during training, we first sample 100k transforms and sort these transforms by their distance to the CIFAR-10-C corruptions. This sorting is done to evenly distribute the augmentations among the 75 (15 corruptions in 5 severities) individual corruptions; *e.g.* the first 75 augmentations in the list are the closest augmentation to each corruption. Then we take a fixed-size subset $\mathbb{A}$ of these transforms and train on augmentations sampled only from this subset using the training parameters from Hendrycks et al. [5]. We select $\mathbb{A}$ three different ways: randomly, taking the $|\mathbb{A}|$ closest augmentations, and taking the $|\mathbb{A}|$ farthest augmentations. We then measure the average corruption error on CIFAR-10-C and plot this error against $|\mathbb{A}|$ in Figure 1.

First, we note that for randomly sampled augmentations, $\mathbb{A}$ does not need to be very large to match AugMix in performance. Even though training on AugMix with our training parameters would normally would produce 5 million uniquely sampled augmentations, only around 1000 are needed to achieve equivalent performance. Training on the closest augmentations exceeds regular AugMix performance with only around 100 unique transforms, which acts as additional evidence that augmentation-corruption similarity correlates with corruption error. This gain in accuracy comes not from having access to better transformations, but from having more frequent access to them at training time. However, the gain is fairly mild at only around 1%, even though the best transformations are sampled all of the time instead of rarely. The gain from frequency is much less than the gain from having more similar augmentations, since choosing the most dissimilar augmentations gives around a 5% drop in accuracy. This suggests that it is a net positive to decrease the frequency of sampling similar augmentations in order to include augmentations similar to another set of corruptions: the gain in accuracy on the new corruption set will likely out weight the small loss in accuracy on the original set.

---

[*]This work completed as part of the Facebook AI residency program.

35th Conference on Neural Information Processing Systems (NeurIPS 2021).

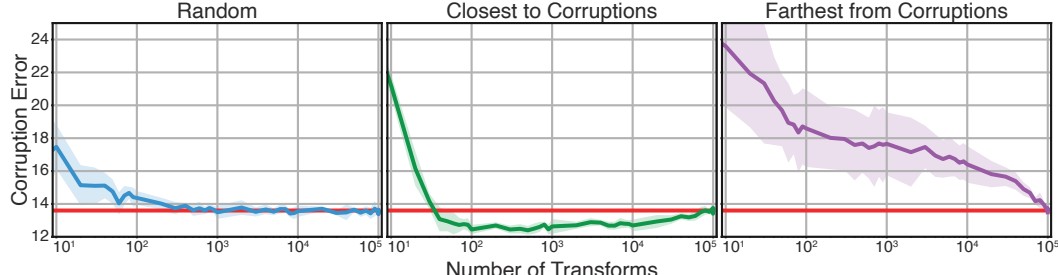

Figure 1: Average corruption error on ImageNet-C as a function the size of a fixed subset of AugMix augmentations. During training, augmentations are only sampled from the subset. The subset is chosen one of three ways: randomly, the most similar augmentations to ImageNet-C, or the least similar augmentations to ImageNet-C. Choosing similar corruptions improves error beyond AugMix, but not by as much that choosing dissimilar augmentations harms it.

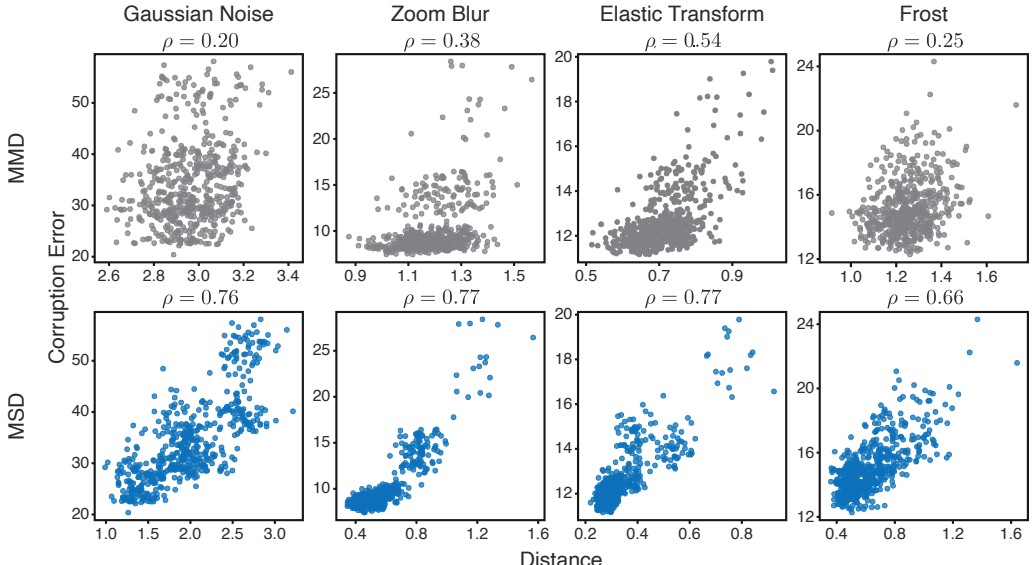

Figure 2: Example relationships between augmentation-corruption distance and corruption error for two distance scores, MMD and MSD. $\rho$ is the Spearman rank correlation. MMD between an augmentation and corruption distribution is not typically predictive of corruption error. MSD correlates well across all four categories of corruption in CIFAR-10-C.

## B Additional MSD and MMD experiments

### B.1 Comparison of MSD and MMD

To support the use of MSD for comparing augmentations and corruptions, we confirm here that the more naive measure of MMD correlates poorly with corruption error. We calculate MMD and MSD as defined in Section 3 of the main text between each augmentation in the augmentation powerset and the corruptions in CIFAR-10-C. Figure 2 shows a comparison of how MMD and MSD correlate with corruption error on sample corruptions. MMD typically shows poor correlation, while MSD has strong correlation in all four categories of corruption.

### B.2 Analyzing generalization with MMD

In Section 3, we argue distributional equivalence is usually not appropriate for studying augmentation-correlation similarity because augmentation distributions are typically broader than any one corruption distribution. However, were an augmentation perceptually similar to a class of corruptions in the

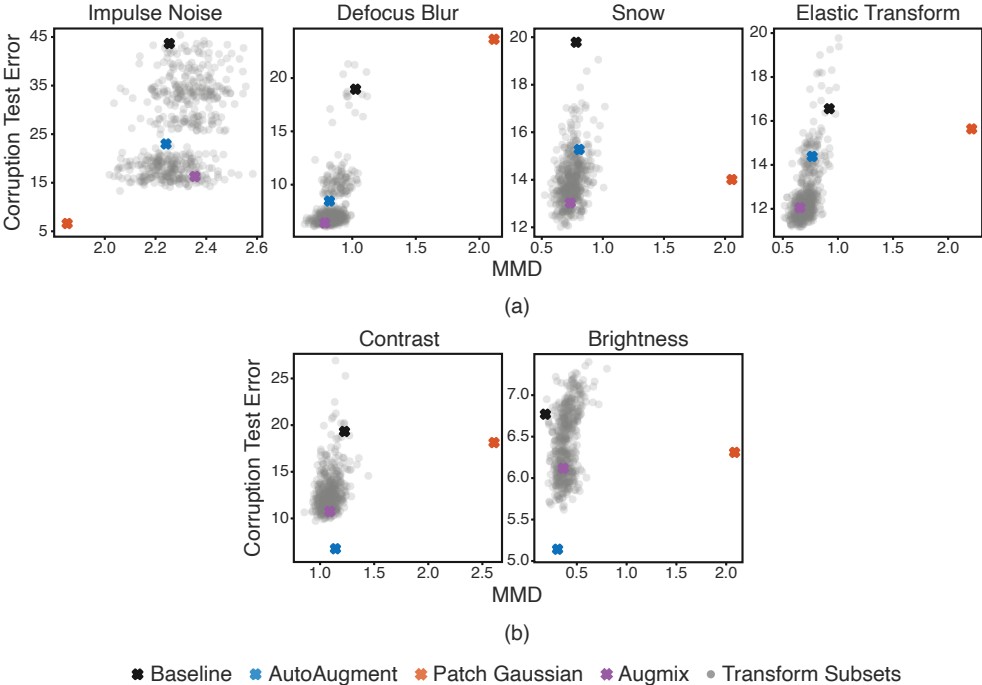

Figure 3: (a) Patch Gaussian shows a low MMD distance on the noise corruptions and a high MMD distance on every other corruption, suggesting that it may be perceptually similar to the noise corruptions in a distributional sense. (b) While AutoAugment contains *contrast* and *brightness* augmentations, it is broad enough that it doesn't have a low MMD to these corruptions. Note that since *brightness* shows poor correlation for MSD, it is possible that in this case the MMD does not change for other reasons.

distributional sense, it might suggest at poor generalization to dissimilar corruptions. Using the simple, necessary but insufficient measure we call MMD in Section 3, we can study a weak sense of distributional equivalence. Figure 3 shows example MMD-error correlations. For Patch Guassian, MMD is low for the noise corruptions and high for everything else, while AutoAugment and AugMix, which are constructed out of many visually distinct transforms, show no strong correlation. This suggests the intuitive result that Patch Gaussian does not just have perceptual overlap with the noise corruptions, but is perceptually similar to them in a more distributional sense. We might then expect poorer generalization from Patch Gaussian to corruptions dissimilar from the noise corruptions, which includes ImageNet-C.

## B.3   MSD vs Batch-Norm Adaptation

It is suggested in Schneider et al. [9] that significant improvement on a set of corruptions may be obtained by adapting only the batch norm parameters of a model trained on clean data to the statistics of the corrupted dataset. One might then expect that there will be a correlation between augmentation-corruption MSD and the error of a model whose batch norm has been adapted to the augmentation distribution. Such a correlation would suggest that a significant benefit of performing augmentations comes from making the batch norm statistics of the training set more similar to the corruption set. Here we test this, performing batch norm adaptation as described in Schneider et al. [9], starting from a model trained with default CIFAR-10 augmentation. We choose the one hyperparameter in their algorithm such that the batch norm parameters are adapted completely to the augmented data distribution. Results are shown in Figure 4. We find that this still correlates well with MSD (though, as is to be expected, less well than training on the augmentations). This lends support to the claim that batch norm statistics are an important aspect of the choice of augmentation.

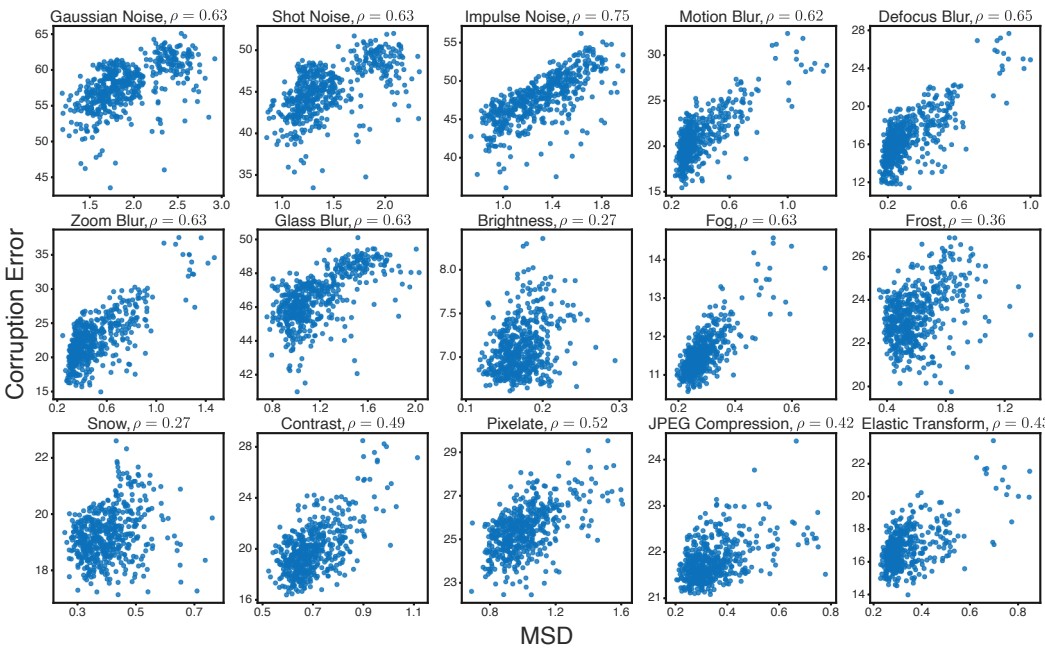

Figure 4: MSD vs. test error on the specified corruption. The test error is obtained by adapting a model's batch norm statistics to the augmented data distribution. The Spearman rank coefficient is given in parenthesis for each corruption. Correlation is still strong but weaker than training the models from scratch on the augmentations.

Table 1: Spearman's rank coefficient for the correlation between MSD and corruption error for two architectures in the feature extractor: WideResNet-40-2 and VGG-19-BN. While WideResNet has slightly better correlations overall, the relative behavior across corruptions remains the same for the two architectures.

| Corruption | WRN | VGG | Corruption | WRN | VGG |
|---|---|---|---|---|---|
| Gaussian Noise | 0.76 | 0.70 | Fog | 0.68 | 0.60 |
| Shot Noise | 0.83 | 0.78 | Frost | 0.66 | 0.66 |
| Impulse Noise | 0.90 | 0.92 | Snow | 0.65 | 0.53 |
| Motion Blur | 0.86 | 0.81 | Contrast | 0.66 | 0.65 |
| Defocus Blur | 0.83 | 0.78 | Pixelate | 0.35 | 0.29 |
| Zoom Noise | 0.77 | 0.68 | JPEG Compression | 0.33 | 0.26 |
| Glass Blur | 0.69 | 0.66 | Elastic Transform | 0.77 | 0.74 |
| Brightness | 0.27 | 0.08 | | | |

# C  MSD Ablation

## C.1  Architecture choice

Here we provide evidence that changing the architecture of the feature extractor used in the definition of MSD does not have any qualitative effect on the correlation with corruption error. We use a version of VGG-19 with batch normalization that has been modified for CIFAR-10. Otherwise, all other parameters are chosen the same. We then repeat the experiment of Section 4. In Table 1 and Figure 5, we show that the qualitative results of this experiment are unchanged when using VGG-19-BN as the feature extractor.

## C.2  Parameter dependencies

In calculating the feature space for transforms and MSD, it is necessary to both pick a number of images to average over and a number of corruptions to average over. In our experiments, we use 100

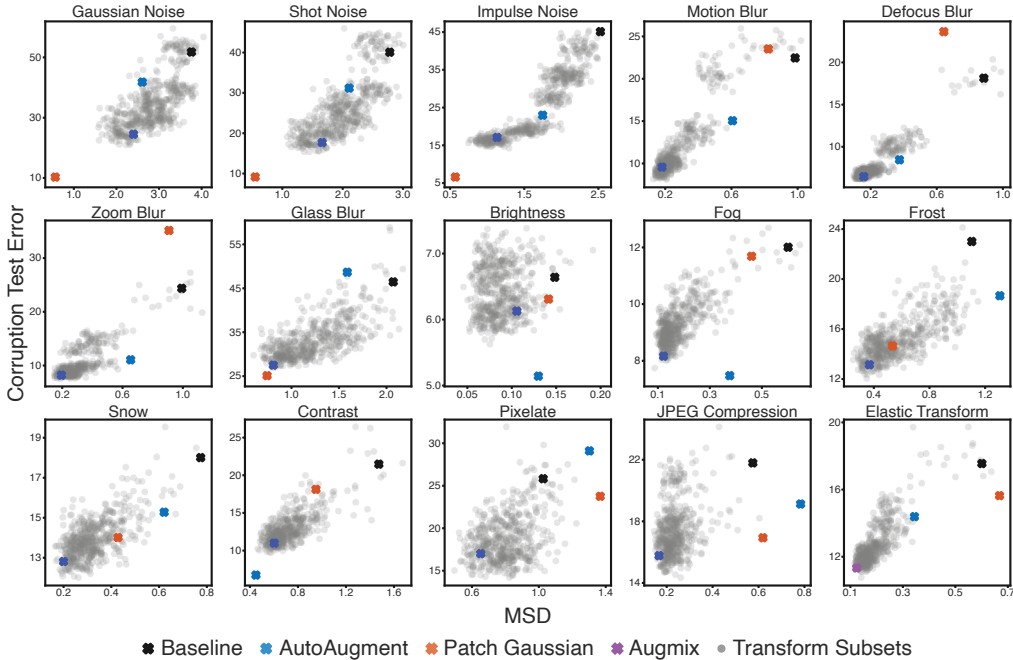

Figure 5: MSD vs corruption test error for which MSD is calculated using VGG-19-BN as the architecture for feature extraction. The corruption error is still calculated using WideResNet-40-2. Compare to Figure 7 to see that the qualitative structure of the correlation is the regardless of which architecture is used for the feature extractor.

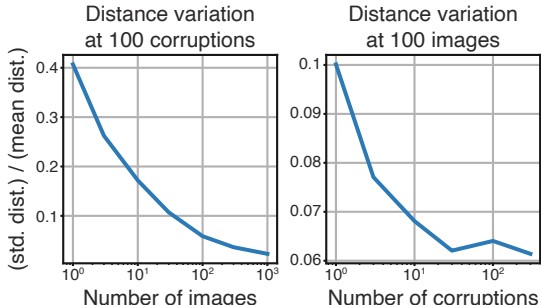

Figure 6: The standard deviation of the distance between an augmentation and a corruption center, taken over 100 resamplings of images and corruptions. The standard deviation is calculated as a percentage of the mean distance, then averaged over 100 augmentation-corruption pairs. At our choice of parameters, 100 images and 100 corruptions, the standard deviation is only around 5% of the distance. This is smaller than the feature size in the scatter plots of Figure 2

images and 100 corruptions. Here we provide evidence that these are reasonable choices for these parameters.

To do so, we use the augmentation scheme from AugMix and corruptions distributions from CIFAR-10-C to randomly sample 100 augmentation-corruption pairs. Then, for different samplings of a fixed number of images and sampled corruptions, we measure the augmentation-corruption distance in the transform feature space 100 times for each augmentation-corruption pair. We calculate the standard deviation of the distance as a percentage of the mean distance for each augmentation-corruption pair, and average this over pairs. The results are shown in Figure 6. For our choice of image and corruption number, the standard deviation in distance is only around 5% of the mean distance, which is smaller than the size of the features in the scatter plots in Figure 2.

# D   ImageNet-$\overline{\text{C}}$ details

## D.1   Dataset construction details

First, 30 new corruptions, examples of which are shown in Figure 9, are adapted from common image filters and noise distributions available online [2, 6]. These corruptions are generated in 10 severities such that the image remains human interpretable at all severities and the distribution of errors on a baseline model roughly matches that of ImageNet-C.

For each corruption, groups of 5 severities are generated that roughly match the average spread in error across severities in ImageNet-C on a baseline model. Seven of these groups are formed for each corruption, each with one of severity 3 through 8 as the center severity of the group of 5.

A candidate dataset is a set of 10 groups of severities, each from a different corruption whose average corruption error on a baseline model is within 1% of ImageNet-C. This is necessary so that a relative decrease in error of data augmented models is normalized against a fixed baseline. Also, more distorted, harder transforms are likely farther away, so if this wasn't fixed maximizing distance would likely just pick the hardest transforms in the highest severities. It was computationally infeasible to enumerate all candidate datasets, so they were sampled as follows. For each choice of 5 corruptions, one choice of severities was selected at random so that the average corruption error was within 1% of ImageNet-C, if it existed. Then random disjoint pairs of two sets of 5 were sampled to generate candidate datasets. 100k candidate datasets are sampled.

Call the set of all corruption-severity pairs in a dataset $\mathbb{C}$. The distance of a candidate dataset to ImageNet-C is defined as

$$d(\mathbb{C}_{\text{new}}, \mathbb{C}_{\text{IN}-\text{C}}) = \mathbb{E}_{c \sim \mathbb{C}_{\text{new}}} \left[ \min_{c' \sim \mathbb{C}_{\text{IN}-\text{C}}} d_{\text{MMD}}(c, c') \right], \tag{1}$$

where $d_{\text{MMD}}$ is defined in Section 3. The minimum helps assure that new corruptions are far from all ImageNet-C corruptions.

This distance is calculated for all 100k sampled candidate datasets. For CIFAR-10, the same parameters described in Section 4 are used to calculate the distance. For ImageNet, the feature extractor is a ResNet-50 trained according to Goyal et al. [3], except color jittering is not used as a data augmentation. Since there is much greater image diversity in ImageNet, we jointly sample 10k images and corruptions instead of independently sampling 100 images and 100 corruptions. Code for measuring distances and training models is based on pyCls [7, 8], and Hydra [10] is used for configuration.

The corruptions are then ranked according the their average contribution to the dataset distance. This entire procedure is repeated 10 times for CIFAR and 5 times for ImageNet, and corruption contributions are averaged. The top 10 are chosen to form the new dataset. These rankings are shown in Figure 7. There may still be multiple candidate datasets made up of these 10 corruptions, differing by the choice of severities. Among these across all runs, we pick the one with error closest to ImageNet-C, though there may still be variation in error run-to-run.

## D.2   Complete results

Here we show results comparing ImageNet/CIFAR-10-C to ImageNet/CIFAR-10-$\overline{\text{C}}$. The 10 transforms chosen for ImageNet-$\overline{\text{C}}$ are blue noise sample (BSmpl), plasma noise (Plsm), checkerboard (Ckbd), cocentric sine waves (CSin), single frequency (SFrq), brown noise (Brwn), perlin noise (Prln), inverse sparkle (ISprk), sparkles (Sprk), and caustic refraction (Rfrac). For CIFAR-10-$\overline{\text{C}}$, there is blue noise sample (BSmpl), brown noise (Brwn), checkerboard (Ckbd), circular motion blur (CBlur), inverse sparkle (ISprk), lines (Line), pinch and twirl (P&T), ripple (Rppl), sparkles (Sprk), and transverse chromatic abberation (TCA). Table 2 compares average results, representing the results from Table 1 in the main text for completeness. A breakdown of ImageNet/CIFAR-10-$\overline{\text{C}}$ results by corruption is in Table 3, including standard deviations for each corruption individually. Stylized-ImageNet is trained jointly with ImageNet for half the epochs, as is done in Geirhos et al. [1]. ImageNet results are averaged over five runs, and CIFAR-10 over ten. For each of the five Stylized-ImageNet runs, we generate a new Stylized-ImageNet dataset using a different random seed and the code provided by Geirhos et al. [1].

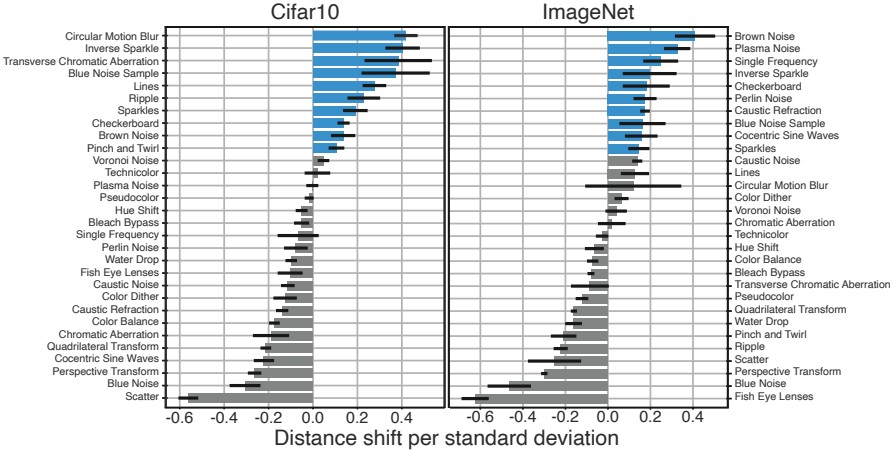

Figure 7: A corruption's average contribution to the distance to ImageNet-C, as a fraction of the population's standard deviation. The blue corruptions are those used to construct ImageNet-$\overline{\text{C}}$.

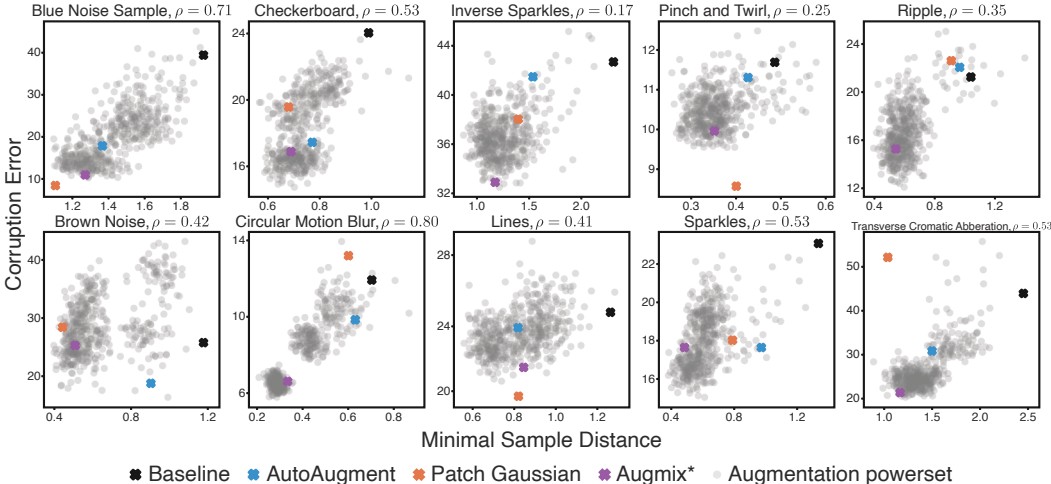

Figure 8: The correlation between MSD and corruption error on the dataset CIFAR-10-$\overline{\text{C}}$. $\rho$ is the Spearman rank correlation.

## D.3   MSD for CIFAR-10-$\overline{\text{C}}$

We repeat the experiment of Section 4 of the main text that measures the correlation between MSD and corruption error using the new corruptions in CIFAR-10-$\overline{\text{C}}$. Results are shown in Figure 8. We find that the correlation is still quite strong for many corruptions, though, like in CIFAR-10-C, there are some corruptions such as *inverse sparkles* where the correlation is weak.

## E   Resource usage

WideResNet-40-2 on CIFAR-10 is trained for about 45 minutes to an hour on 1 V100 GPU, while ResNet-50 on ImageNet is trained for approximately 20 hours on 8 V100 GPUS. Collecting augmentation features for MSD requires 45 to an hour on 1 V100 GPU. In-memory corruption evaluation and feature extraction for CIFAR-10/ImageNet-C and the newly introduced corruptions is often CPU limited and runtimes vary significantly from corruption type to corruption type. This ranges up to approximately 6 hours on 80 Intel Xenon 2.2Ghz CPUs for per corruption and severity for ImageNet, or up to approximately 8 minutes per corruption and severity on 40 CPUs for CIFAR-10. When calculating distances for choosing CIFAR-10/ImageNet-$\overline{\text{C}}$, CIFAR-10 uses the same amount of time

Table 2: Comparison between performance on ImageNet/CIFAR10-C and ImageNet/CIFAR10-$\overline{\text{C}}$. Standard deviations are over 10 runs for CIFAR-10 and 5 runs for ImageNet. *ANT, DeepAugment (DA), and DeepAugment+AugMix (DA+AM) results use the pre-trained models provided with the respective papers and thus have different training parameters and only one run.

|  | **IN-C** | **IN-$\overline{\text{C}}$** |  |
| --- | --- | --- | --- |
| Aug | Err | Err | $\Delta$IN-C |
| Baseline | $58.1_{\pm0.4}$ | $57.7_{\pm0.2}$ | -0.5 |
| AA | $55.0_{\pm0.2}$ | $55.7_{\pm0.3}$ | +0.7 |
| SIN | $52.4_{\pm0.1}$ | $55.8_{\pm0.3}$ | +3.4 |
| AugMix | $49.2_{\pm0.7}$ | $52.4_{\pm0.2}$ | +3.2 |
| PG | $49.3_{\pm0.2}$ | $56.6_{\pm0.4}$ | +7.3 |
| ANT* | 48.8 | 53.9 | +5.1 |
| DA* | 46.6 | 51.0 | +4.4 |
| DA+AM* | 41.0 | 48.3 | +7.3 |

|  | **C10-C** | **C10-$\overline{\text{C}}$** |  |
| --- | --- | --- | --- |
| Aug | Err | Err | $\Delta$C10-C |
| Baseline | $27.0_{\pm0.6}$ | $27.1_{\pm0.5}$ | +0.1 |
| AA | $19.4_{\pm0.2}$ | $21.0_{\pm0.4}$ | +1.6 |
| AugMix | $11.1_{\pm0.2}$ | $16.0_{\pm0.3}$ | +4.9 |
| PG | $17.0_{\pm0.4}$ | $23.8_{\pm0.5}$ | +6.8 |

Table 3: Breakdown of performance on individual corruptions in ImageNet/CIFAR10-$\overline{\text{C}}$. Standard deviations are over 10 runs for CIFAR-10 and 5 runs for ImageNet. Examples and full names of each corruption are given in Appendix F. *ANT, DeepAugment (DA), and DeepAugment+AugMix (DA+AM) results use the pre-trained models provided with the respective papers and thus have different training parameters and only one run.

**ImageNet-$\overline{\text{C}}$ Corruptions**

| Aug | BSmpl | Plsm | Ckbd | CSin | SFrq | Brwn | Prln | ISprk | Sprk | Rfrac |
| --- | --- | --- | --- | --- | --- | --- | --- | --- | --- | --- |
| Baseline | $68.6_{\pm0.5}$ | $71.7_{\pm0.7}$ | $49.4_{\pm0.6}$ | $84.7_{\pm0.5}$ | $79.0_{\pm0.8}$ | $37.5_{\pm0.5}$ | $34.3_{\pm0.1}$ | $32.4_{\pm0.5}$ | $76.7_{\pm0.2}$ | $42.8_{\pm0.2}$ |
| AA | $54.8_{\pm0.7}$ | $68.3_{\pm0.7}$ | $43.8_{\pm1.0}$ | $86.5_{\pm0.6}$ | $78.8_{\pm0.9}$ | $34.5_{\pm0.8}$ | $33.8_{\pm0.2}$ | $36.1_{\pm1.0}$ | $77.1_{\pm1.2}$ | $43.8_{\pm0.2}$ |
| SIN | $54.7_{\pm1.5}$ | $69.8_{\pm1.1}$ | $52.8_{\pm1.0}$ | $79.6_{\pm0.4}$ | $69.2_{\pm0.6}$ | $37.8_{\pm0.4}$ | $35.3_{\pm0.1}$ | $37.0_{\pm0.5}$ | $77.3_{\pm0.8}$ | $44.1_{\pm0.2}$ |
| AugMix | $43.2_{\pm0.8}$ | $72.2_{\pm0.4}$ | $46.1_{\pm0.2}$ | $76.3_{\pm0.3}$ | $67.4_{\pm0.7}$ | $38.8_{\pm0.5}$ | $32.4_{\pm0.1}$ | $32.3_{\pm0.2}$ | $76.4_{\pm0.4}$ | $39.2_{\pm0.2}$ |
| PG | $60.3_{\pm2.9}$ | $74.1_{\pm0.7}$ | $48.5_{\pm1.0}$ | $82.1_{\pm0.4}$ | $76.7_{\pm0.8}$ | $38.9_{\pm0.4}$ | $34.6_{\pm0.1}$ | $32.1_{\pm0.7}$ | $76.5_{\pm0.6}$ | $42.1_{\pm0.4}$ |
| ANT* | 35.8 | 75.5 | 56.9 | 76.4 | 63.7 | 41.0 | 35.2 | 35.0 | 76.1 | 43.3 |
| DA* | 41.7 | 73.3 | 53.9 | 74.6 | 50.9 | 37.2 | 30.3 | 32.9 | 74.7 | 40.9 |
| DA+AM* | 34.9 | 67.9 | 49.8 | 69.7 | 48.0 | 35.2 | 30.6 | 32.9 | 74.3 | 39.8 |

**CIFAR-10-$\overline{\text{C}}$ Corruptions**

| Aug | BSmpl | Brwn | Ckbd | CBlur | ISprk | Line | P&T | Rppl | Sprk | TCA |
| --- | --- | --- | --- | --- | --- | --- | --- | --- | --- | --- |
| Baseline | $42.9_{\pm5.1}$ | $27.2_{\pm0.5}$ | $23.3_{\pm0.6}$ | $11.8_{\pm0.4}$ | $43.3_{\pm0.8}$ | $26.2_{\pm0.9}$ | $11.3_{\pm0.3}$ | $21.6_{\pm1.2}$ | $21.0_{\pm1.1}$ | $42.9_{\pm2.7}$ |
| AA | $17.7_{\pm1.7}$ | $17.5_{\pm0.5}$ | $17.6_{\pm0.5}$ | $9.5_{\pm0.3}$ | $40.4_{\pm1.5}$ | $23.6_{\pm0.7}$ | $10.7_{\pm0.3}$ | $23.5_{\pm0.5}$ | $17.5_{\pm0.7}$ | $31.8_{\pm1.8}$ |
| AugMix | $9.8_{\pm0.7}$ | $27.8_{\pm1.3}$ | $13.4_{\pm0.4}$ | $5.9_{\pm0.2}$ | $30.3_{\pm0.7}$ | $18.0_{\pm0.6}$ | $8.3_{\pm0.2}$ | $12.1_{\pm0.4}$ | $15.5_{\pm0.5}$ | $19.2_{\pm1.0}$ |
| PG | $9.0_{\pm1.1}$ | $30.1_{\pm1.1}$ | $21.6_{\pm0.8}$ | $12.8_{\pm0.5}$ | $35.4_{\pm1.6}$ | $20.6_{\pm0.5}$ | $8.8_{\pm0.2}$ | $21.5_{\pm0.9}$ | $19.3_{\pm0.5}$ | $59.5_{\pm3.5}$ |

per corruption as evaluation of the corruption, while ImageNet uses 1/5th the time, simply as a result of the number of images processed in each case.

# F  Glossary of transforms

This appendix contains examples of the augmentations and corruptions discussed in the text. Figure 9 shows the 30 new corruptions introduced in Section 5. These transforms are adapted from common online filters and noise sources [2, 6]. They are designed to be human interpretable and cover a wide range transforms, including noise additions, obscuring, warping, and color shifts.

Figure 10 shows the 9 base transforms used to build augmentation schemes in the analysis. These are transforms from the Pillow Image Library that are often used as data augmentation. They have no exact overlap with either the corruptions of ImageNet-C or the new corruptions we introduce here. There are five geometric transforms (shear x/y, translate x/y, and rotate) and four color transforms (solarize, equalize, autocontrast, and posterize). We choose this particular set of augmentations following Hendrycks et al. [5].

Figure 11 shows example corruptions from the ImageNet-C benchmark [4]. They a grouped into four categories: noise (gaussian noise, shot noise, and impulse noise), blurs (motion blur, defocus blur, zoom blur, and glass blur), synthetic weather effects (brightness, fog, frost, and snow), and digital transforms (contrast, pixelate, JPEG compression, and elastic transform).

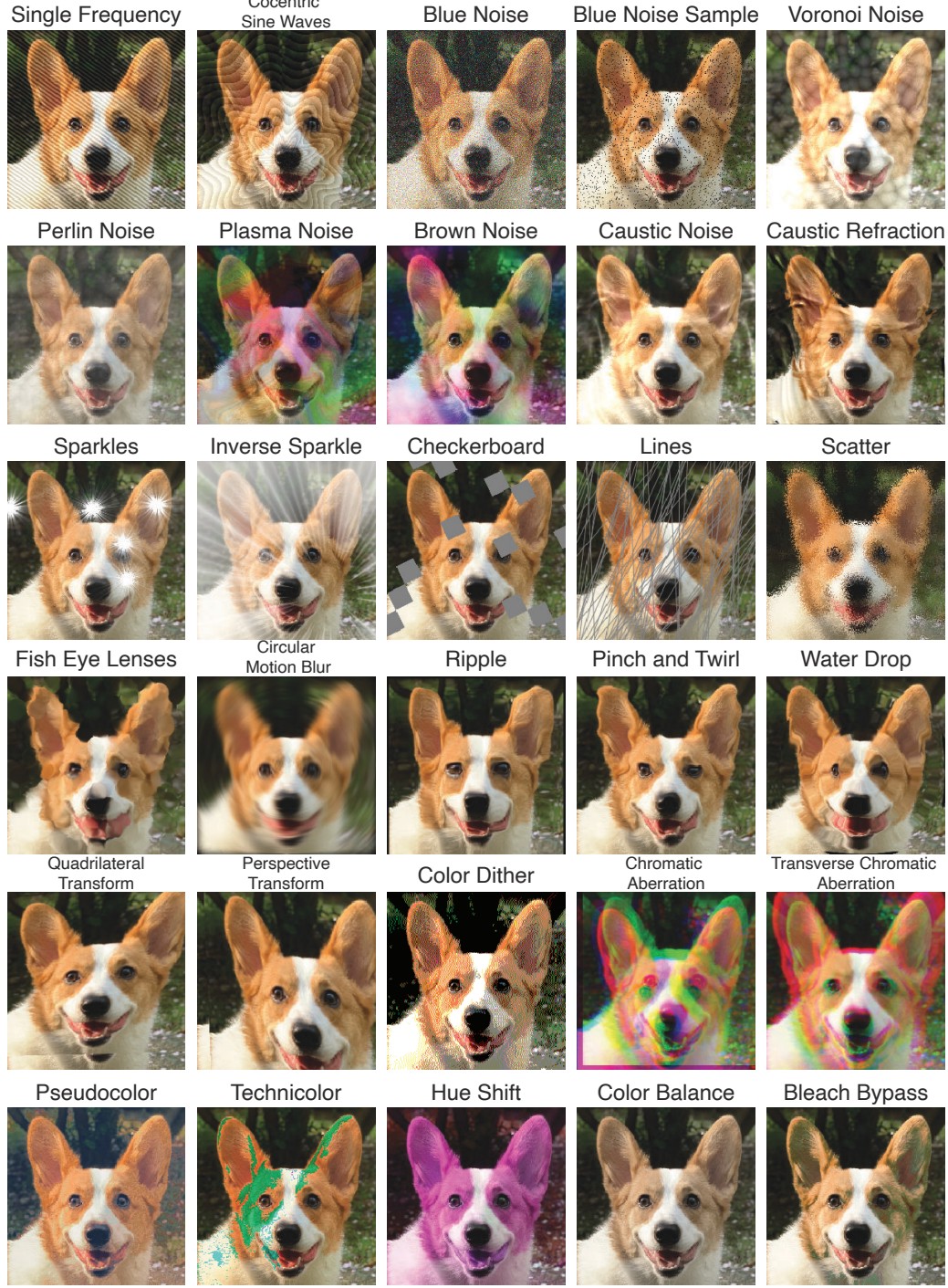

Figure 9: Examples of each corruption considered when building the dataset dissimilar to ImageNet-C. Base image © Sehee Park.

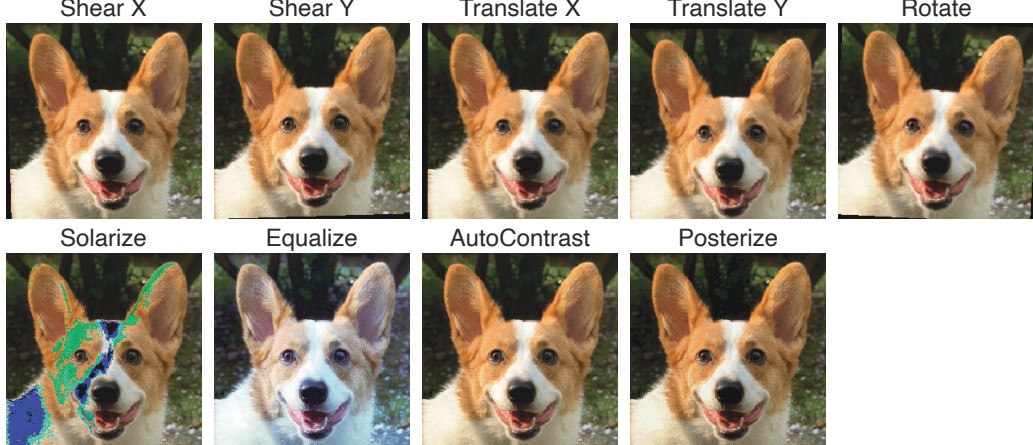

Figure 10: The nine base transforms used as augmentations in analysis. Base image © Sehee Park.

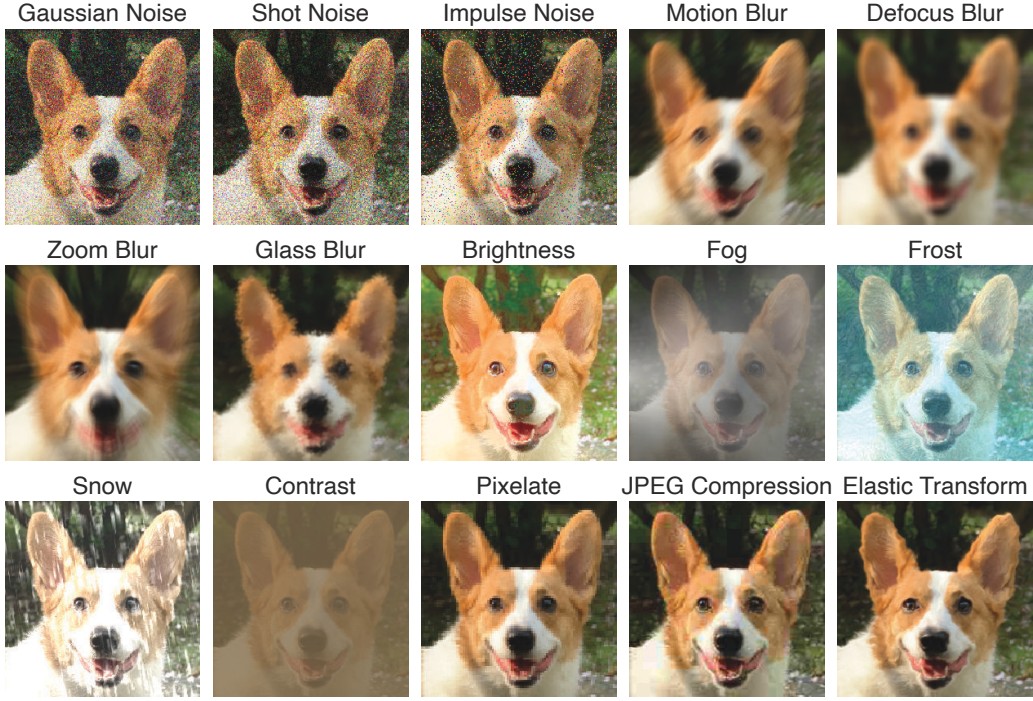

Figure 11: Examples of the 15 corruptions in the ImageNet-C corruption benchmark [4]. Base image © Sehee Park.