# OpenReview forum: "On Interaction Between Augmentations and Corruptions in Natural Corruption Robustness"
_NeurIPS.cc/2021/Conference — NeurIPS 2021 Poster_

### Official Review · Reviewer_89ct · 2021-07-16

**Rating:** 7
**Confidence:** 4

**Summary:**

The paper study the similarities between the augmentations/augmentation schemes with the corruptions and reason that the perceptual similarity between them is a strong predictor for the robustness improvements. The authors propose a new metric, Minimal Sample Distance (MSD), to evaluate the perceptual similarity between a corruption and an augmentation statistically. In addition, authors construct a new corruption benchmark ImageNet-C bar that is perceptually dissimilar to ImageNet-C. Results on this benchmark show that recent robust models with different augmentations schemes have reduced performance than ImageNet-C, suggesting that these augmentation schemes share perceptual similarity to ImageNet-C.

**Limitations And Societal Impact:**

I appreciate the authors for addressing the limitations and societal impacts of their work..

**Main Review:**

Originality:
The paper provides a new interesting finding and the metric proposed to evaluate the perceptual similarity between corruptions and augmentations is novel. Moreover, the paper also proposes a new corruption benchmark ImageNet-C bar.

Quality:
The paper is technically sound and the claims like “Augmentation-corruption perceptual similarity is a strong predictor of corruption error” is moderately supported based on the results from Table 1.

Clarity:
The paper is clearly written, well organized and included important details to avoid possible misunderstandings.

Significance:
The paper studies an important relationship between the recent data augmentation schemes and the robustness of common corruptions. Future works would benefit from the interesting findings and the benchmark proposed in this paper. To my knowledge, this is the first paper that proposes a statistical measure to find the correlation between augmentations and the corruptions.
Results from Table 1 show that recent robust models drop few percent on ImageNet-C bar compared to ImageNet-C. It is not trivial to conclude or not easy to support the authors claim “Augmentation-corruption perceptual similarity is a strong predictor of corruption error” based on the few percent drop from this table. However, individual corruption errors reveals that performance on some of the corruptions are worse than the others.

ImageNet-C bar is constructed to be perceptually dissimilar to ImageNet-C in the transform feature space. Are any of the ImageNet-C bar corruptions perceptual to existing data augmentation schemes?  For example, any lower MSD between augmentations from Augmix (or other augmentation schemes) and ImageNet-C bar? Does that explain augmentation schemes improvements on some of the corruptions in ImageNet-C bar? Illustration similar to Figure 5 on ImageNet-C bar would be helpful here.

Does AugMix tested with Jensen-Shannon divergence loss? If so, it would be interesting to check the AugMix results with and without divergence loss on ImageNet-C bar.

How does the augmentations generated by a deep network DeepAugment (Hendrycks et al. [15]) would correlate with the ImageNet-C and ImageNet-C bar corruptions? How does DeepAugment, DeepAugment+Augmix would work on ImageNet-C bar?

Overall, I find this paper interesting and accept it as Good paper. However, I would reconsider my rating based on other reviews and author responses.

-------------------------------------------------------
Post rebuttal remarks:

I read the other reviews and authors responses. I thank the authors for providing additional insights. Given the rebuttal time, I understand that some of posed questions might not be completely addressed. However, I highly encourage the authors to address those questions in their final version, as it is beneficial to the community. Finally, I believe that this work is a significant contribution to the community and tend to keep my rating to accept it.


**Time Spent Reviewing:**

5

---

> ### Author Response · Authors · 2021-08-10
> **Author response to reviewer 89ct**
>
> Thank you for taking the time to review our paper and providing excellent comments and questions. We are glad you found it easy to follow and agreed that we present technically sound results that provide the first statistical measure to find the correlation between augmentations and corruptions.
>
> **ImageNet-C-bar correlation with augmentations.** We agree that a plot similar to Figure 5 for ImageNet-C or ImageNet-C-bar would provide additional insight. However, this is unfortunately prohibitively expensive since each point in a plot like Figure 5 would be a different fully trained ImageNet model. We believe our results can generalize to ImageNet. We can instead create a plot similar to Figure 5 for CIFAR-10-C-bar, provided on the anonymized google drive here: https://drive.google.com/file/d/1sxt8nu9wGK1Ek3PSyE0lDCOT8YMF61xq/view?usp=sharing. We see that the correlation between MSD and corruption error still holds for many corruptions, though on average it is somewhat weaker than for CIFAR-10.
>
> In addition we can provide a few results for ImageNet-C/C-bar. The average MSD for AugMix to ImageNet-C is 5.9, and to ImageNet-C-bar is 7.1, while for Patch Gaussian it is 5.4 and 7.5 for ImageNet-C and ImageNet-C-bar respectively. So for both these augmentations, ImageNet-C-bar is farther away than ImageNet-C. Additionally, consistent with their errors on the two datasets, Patch Gaussian is closer to ImageNet-C but farther from ImageNet-C-bar than AugMix. This pattern is not perfect: Stylized-ImageNet (10.1 MSD to ImageNet-C, 10.8 MSD to ImageNet-C-bar) is far from both yet performs better than Patch Gassian on ImageNet-C-bar; we discuss in the text how Stylized-ImageNet may differ in behavior from other augmentation schemes.
>
> **AugMix with JSD loss.** The test on AugMix is performed using the Jensen-Shannon divergence loss.  We agree that AugMix without the JSD loss may also be interesting; however, in Table 1 we wished to compare several augmentation schemes as they are presented in the literature in order to show inconsistent behavior on dissimilar corruptions affects real-world robust augmentations. Since the JSD loss is part of the primary recipe presented in Augmix [14], we use it here as well.
>
> **DeepAugment.** We agree DeepAugment is another augmentation scheme that could be interesting to study. Unfortunately, due to the complexity and offline nature of the DeepAugment augmentation scheme, it is not straightforward to calculate its correlation with ImageNet-C/C-bar given the limited time for rebuttal and discussion. However, below we provide comparisons of the performance of DeepAugment and DeepAugment+Augmix on ImageNet-C/C-bar using the pre-trained models provided by their paper. While both perform quite well on average, they show large discrepancies across different corruptions in ImageNet-C-bar (e.g., a ~40% improvement on single frequency noise but a ~2% improvement on inverse sparkles), consistent with our claims and with the behaviors of the other augmentations.
>
> Augmentation | IN-C | IN-Cbar | Delta | BSmpl |Plsm |Ckbd |CSin |SFrq |Brwn |Prln |Sprk |ISprk |Rfrac
> -------------------|--------|------------|---------|----------|--------|--------|-------|-------|--------|------|-------|-------|-----
> DeepAugment| 46.6  | 51.0   |   +4.4 |  41.7  |   73.3  |53.9 |74.6  |50.9  |37.2 | 30.3 |32.9 |74.7   |40.9
> DA+Augmix    | 41.0  | 48.3   |   +7.3 |  34.9  |   67.9  |49.8 |69.7  |48.0 | 35.2  |30.6 |32.9 |74.3   |39.8
>
> **References:**
>
> [14] Hendrycks, D., Mu, N., Cubuk, E. D., Zoph, B., Gilmer, J., and Lakshminarayanan, B. AugMix: A simple data processing method to improve robustness and uncertainty. In ICLR, 2019.

---

### Official Review · Reviewer_Wm6Y · 2021-07-17

**Rating:** 7
**Confidence:** 5

**Summary:**

The paper introduces a metric, the Minimal Sample Distance (MDS), that quantifies the distance between the distribution of samples augmented with a certain augmentation and samples affected by a certain corruption. The authors show a correlation between the MDS and the corruption error, and use this metric to show that existing corruption benchmarks, e.g. ImageNet-C, are not broad enough to properly evaluate the robustness of image classifiers. Thus, they expand ImageNet-C with further corruptions that are not perceptually similar to those already available.

**Limitations And Societal Impact:**

The author discuss limitations and potential societal impact in a broad sense, being the work addressing a general problem with large potential impact. However, I did not see explained how this work can be used to prevent issues and failure cases, as the authors only mention this possibilities briefly.

**Main Review:**

The paper addresses an important problem of Computer Vision, namely the robustness of image classifiers to common test-time corruptions that are not seen during training. This links with the generalization capabilities of classifiers, and their applicability to real-world problems.

Strengths:
- Extension of the ImageNet-C benchmark with perceptually-dissimilar corruptions
- A metric (MSD) that measures the relation between augmentation and test-time corruptions
- Demonstration that the robustness of computer vision models needs to be addressed with a broader approach


Concerns:
- By the definition of the MSD, it tries to measure how much a certain augmentation is contained in the distribution of a given corruption, by computing the distance between the nearest sampled point. I have the impression that such evaluation might not be stable, having some flaws due to the randomness of agumentation and corruptions. It seems to me that the overlapping area between the two distributions might be a better option. I did not find a possible intepretation of this type, and would like the authors to comment about.
- 'perceptual' similarity is not defined precisely. This opens room for doubts and interpretations. When, precisely, two corruptions can be said perceptually similar? and what are the conditions that determine that two augmentations/corruptions are not similar?
- I have the impression that a lot of the similarity/dissimilarity can be explained by looking at the spatial frequency components that certain agumentations and corruptions affect. Using similar agumentations (likely meaning that they inject or modify similar spatial frequency statistics) is clearly not much beneficial, especially when compared to using augmentations that affect a larger portion of the spectrum, thus more dissimilar. More concrete explanations of the meaning of perceptual similarity/dissimilarity should be provided, in terms of frequency analysis or other approaches that the authors might find more suitable.
- the paper and results discussion rely too much on the supplementary material. The paper is thus not self-contained and the observation and delivered messages are not strong enough without looking into extra materials.

I think this paper addresses an important problem and proposes data and metrics to stimulate advancements and new solutions in the field. However, some results and interpretations need a little clarrification.

**Time Spent Reviewing:**

5

---

> ### Author Response · Authors · 2021-08-10
> **Author response to reviewer Wm6Y**
>
> Thank you for taking the time to review our paper and for providing excellent comments and concerns. We are glad you agree this paper addresses an important problem in the field and that MSD and ImageNet-C-bar help advance further research and solutions.
>
> **Distribution overlap.** During development of the paper we considered using the KL divergence as the tool to measure augmentation-corruption similarity. This is a distribution overlap measure that, like MSD, is asymmetric in the necessary way to compare broad augmentation schemes to individual corruptions. However, the KL divergence is intractable and must instead be estimated by using a trained classifier or other means. A version of this design had a similar correlation with corruption error as MSD (the average Spearman correlation for all corruptions was 0.64, plots for all corruptions are provided at this anonymized google drive link: https://drive.google.com/file/d/1-QluZurWEVxyc0PVXDJoBCm1V20ly_qw/view?usp=sharing). However, the introduction of a classifier significantly complicated the design of the measure and added many additional parameters to tune. We settled on MSD as a measure since it was significantly simpler to design and implement. Empirically we have not found stability problems with MSD: the results of Section 4 continue to hold when the experiment is repeated, and we have discussed stability with respect to parameter dependency in Appendix E.2.
>
> **Definition of perceptual similarity.** We understand your concern regarding a precise definition of ‘perceptual similarity’. Indeed, inconsistent treatment of similarity between augmentations and corruptions is the original motivation for this paper. We found that when an augmentation was not algorithmically identical to a corruption but produced visually similar results, it would be excluded when testing corruption robustness sometimes but not other times. This made it hard to fairly compare robustness methods. The purpose of our paper is to provide a quantifiable statistical measure that captures similarity in this sense. We aim to justify the value of our measure empirically instead of from first principles; empirical measures that use a neural network as a proxy for ‘perceptual’ have prior precedent [39]. This is done by showing that MSD does correlate with corruption test error, and that this correlation matters: the performance of an augmentation on corruptions dissimilar to ImageNet-C is not consistent, so comparing augmentation schemes with ImageNet-C alone may not be sufficient.
>
> We expect similarity in the spectra between augmentations and corruptions is a complementary correlation to the one we consider here. As an example: in the context of Section 4, the geometric augmentations both have lower MSD and better test performance on both the ‘contrast’ corruption and the blur corruptions. This is despite the fact that the blur corruptions are low pass filters (and thus modify the high frequencies), yet ‘contrast’ is a very low frequency distortion. The spectra of the corruptions are certainly also an important aspect of better understanding corruption robustness. For example, we discussed a relevant work [35] in the related work section.
>
> **Presentation/Misc.** We apologize that you did not find the paper self-contained. We will try to improve the presentation and make the paper more self-contained in the next version. We will also work to improve the discussion involving failure cases and societal impact.
>
> **References:**
>
> [39] Zhang, R., Isola, P., Efros, A. A., Shechtman, E., and Wang, O. The unreasonable effectiveness of deep features as a perceptual metric. In CVPR, 2018.
>
> [35] Yin, D., Lopes, R. G., Shlens, J., Cubuk, E. D., and Gilmer, J. A Fourier perspective on model 419 robustness in computer vision. In NeurIPS, 2019.

---

### Official Review · Reviewer_uoqv · 2021-07-21

**Rating:** 7
**Confidence:** 4

**Summary:**

In this paper, the authors propose an exciting perspective to rethink the interaction between augmentation and corruption. The perceptually similar augmented training images could be helpful for better performance on common natural corrupted test-time images. The creation of CIFAR/ImageNet-$\overline{C}$ and the related discussion (section 6) indicates the future direction of data augmentation for corruption robustness with enough care in evaluation protocol.

**Ethics Review Area:**

["I don’t know"]

**Limitations And Societal Impact:**

Besides train-time augmentation, there are a bunch of methods that leverage the test-time/source-free adaptation[1][2][3] techniques to update the model during the inference phase. I appreciate the authors' efforts in the existing evaluation. I wonder if these methods could also be evaluated as the representative approaches for the group of test-time dynamic models.

- [1] Sun, Yu, et al. "Test-time training with self-supervision for generalization under distribution shifts." International Conference on Machine Learning. PMLR, 2020.
- [2] Liang, Jian, Dapeng Hu, and Jiashi Feng. "Do we really need to access the source data? source hypothesis transfer for unsupervised domain adaptation." International Conference on Machine Learning. PMLR, 2020.
- [3] Wang, Dequan, et al. "Tent: Fully Test-Time Adaptation by Entropy Minimization." International Conference on Learning Representations. 2020.

**Main Review:**

- The proposed method is easy to follow. The overall writing looks well-organized to me. Both big picture and implementation details are discussed intensively. I believe the readers could catch up with the most critical observations in an easy way.
- The illustrations are helpful for the readers to realize the main motivations and contributions of the proposed method. Specifically, the correlation plots between minimum sample distance (MSD) and corruption error look well-structured to highlight the key points.
- The contributions look solid to me: 1) perceptual similarity measurement, minimal sample distance (MSD); 2) new dataset, CIFAR/ImageNet-$\overline{C}$, perceptually dissimilar training samples; 3) understanding and analysis on the interaction and the discussion on the future direction.

**Time Spent Reviewing:**

4

---

> ### Author Response · Authors · 2021-08-10
> **Author response to reviewer uoqv**
>
> Thank you for taking the time to review our paper and for providing excellent questions and comments. We are glad you found the paper well-organized and agree that MSD and CIFAR/ImageNet-C-bar are solid contributions to the field. Thank you in particular for noting our discussions on the big picture and future directions -- we think this is a valuable component of advancing understanding and further research.
>
> The application of our methods to test-time dynamic models is an interesting idea for future study; thank you for raising the idea. The context is different (i.e. generalization to unknown corruptions is not the goal, since the corruptions are available for test-time training) and some details differ (e.g. our exact design here requires augmentations and corruptions to be applied to the same set of underlying clean images, which is unlikely to be an option in the test-time training case).  As such it would take some further thinking to apply our methods; however, the general idea of MSD might still be useful as a way to compare a broad distribution to a set of narrow ones. As one possible example, in an online test-time training regime MSD could perhaps be used to evaluate how similar past test-time corruptions are to future test-time corruptions. We will add relevant citations and discussion of test-time dynamic models to the next version.

---

### Decision · Program_Chairs · 2021-09-27

**Decision:**

Accept (Poster)

**Comment:**

This is an interesting paper, and there was clear agreement among the reviewers that this marks an interesting contribution.